# The Relationship Between Willingness to Forgive and Health-Related Quality of Life in Older Adults in Portugal and Spain

**DOI:** 10.3390/geriatrics10040090

**Published:** 2025-07-02

**Authors:** Cristiane Pavanello Rodrigues Silva, Fausto J. Barbero-Iglesias, Luis Polo-Ferrero, José I. Recio-Rodríguez

**Affiliations:** 1RISE-Health, Escola Superior de Saúde de Santa Maria, Tv. de Antero de Quental 173 175, 4049-024 Porto, Portugal; cristiane.silva@santamariasaude.pt; 2Department of Nursing and Physiotherapy, Institute of Biomedical Research of Salamanca, University of Salamanca, 37007 Salamanca, Spain; fausbar@usal.es (F.J.B.-I.); pfluis@usal.es (L.P.-F.); 3Primary Care Research Unit of Salamanca, Research Network on Chronicity, Primary Care and Health Promotion, 37007 Salamanca, Spain

**Keywords:** health-related quality of life, older individuals, forgiveness

## Abstract

Objectives: To describe, understand, and correlate willingness to forgive with self-perceived health-related quality of life, including the components of quality of physical health and mental health. Methods: Conducted with 30 older individuals, ≥65 years old, with preserved cognitive abilities, literacy ≥ four years of education, living in Portugal and Spain. The 12-Item Short-Form Health Survey (SF-12) was used to assess self-perceived health-related quality of life, and the Heartland Forgiveness Scale (HFS) was used to measure willingness to forgive. Results: There was a direct positive correlation between willingness to forgive and perception of health-related quality of life, especially for the mental health component, educational level, cohabitation, and work activity. Conclusions: Forgiveness could play a significant role in the emotional health and quality of life of the elderly. Strategies to develop forgiveness can benefit the active aging process, contributing to improved health-related quality of life in older individuals.

## 1. Introduction

Forgiveness has different and subjective meanings that vary among many factors according to age, culture, values, beliefs, and time, regardless of the associated evidence and concrete facts, because forgiveness is accompanied by the “attitude of forgiving”, which is interpersonal, according to the judgment made by the person who was “attacked”, and may or may not be granted to the “guilty one”, according to the individual’s disposition to forgive. Forgiveness is a complex process that is not merely “sentimental”, passing through the conscious decision to forgive [1,2].

The approach to forgiveness as a science began in the therapeutic community after the 1984 publication of “Forgive and Forget: Healing the Hurts We Don’t Deserve” by Lewis Smedes, a theologian who started the movement with the idea that forgiveness can be beneficial for physical and mental health [1]. Robert Enright pioneered the idea of the reasoning skills of forgiveness in parallel with the development of skills about justice, which had been developed by psychologist Lawrence Kohlberg (1984). Enright defends the integration between behavior, judgment, and affect, with the replacement of negative thoughts, actions, and feelings by more positive thoughts, actions, and feelings; he understands forgiveness as a complex process, a triad that involves three aspects: forgiving, receiving forgiveness, and self-forgiveness [1].

According to the science of forgiveness, there is a paradox in this process, because as we reach out to those who have hurt us, we are the “healed” [2]. In this sense, the work of developing the capacity to forgive flourishes in order to improve the sustainable health and quality of life of the human person, with an impact on the improvement of their and others’ health-related quality of life [2,3]. Forgiveness behavior can vary according to how the injustices suffered are perceived, which varies in different cultures, and in different social, family, and work relationships, as well as in different stages of life [2].

Forgiveness is an option, a process that involves cognitive, emotional-affective, behavioral, psychosocial, and interpersonal dimensions; it expresses an evolutionary extension and personal maturity—with no possibility of modifying the offenses that happened in the past, but with the possibility of reducing anxiety, stress and depression—functioning as a protective mechanism, especially of mental health, both in the present and in the future [2,3,4,5]. For example, for self-forgiveness, which may depend less on the behavior of others, studies indicate that an individual who experiences self-forgiveness improves their psychological health, including their satisfaction with life and self-esteem, in addition to experiencing emotional stability and self-perceived quality of life [2,3,4,5].

Efforts have been made to promote forgiveness interventions in the population of children, adolescents, and adults with various physical and/or mental health problems, as well as couples, especially in situations of violence and abuse, with benefits in personal health and quality of life and the maintenance of positive relationships. However, little is known about these interventions in older individuals; this approach is extremely important, as transgressions and offenses that are poorly resolved throughout life can contribute to symptoms of depression and social isolation in this population [6,7,8].

A study conducted by Ingersoll-Dayton and Krause (2010) [8] indicates that there is a relationship between the elderly’s perception of “God’s Unforgiveness” and “Others’ Unforgiveness” and symptoms of depression, which seems to be transmitted primarily by “Not Forgiving Oneself” and the failure of a mechanism that elderly people use (rumination).

The attitude of forgiveness can positively influence the quality of aging and can be developed or improved with the help of professional interventions, although there is a need for a greater number of studies evaluating forgiveness and its associations with physical and emotional health and quality of life, which can have a direct or indirect impact on the health-related quality of life of older individuals [9]. However, through the evaluation of published studies, it is already possible to affirm that forgiveness, especially of oneself, directly, and then of others, indirectly, can signal a greater capacity to fully live in aging, and older individuals who forgive more are subject to a better health-related quality of life [9].

The global aging population is recognized by the World Health Organization (WHO) with a clear concern about their health, including the publication of the Global Strategy and Action Plan for Aging and Health for 2016–2020 and meeting the Sustainable Development Goal (SDG) of Health and Well-Being [10]. According to an OECD (Organization for Economic Co-operation and Development) report, the countries that are currently aging the fastest are Greece, Korea, Poland, Portugal, Slovenia, and Spain [11]. Therefore, any and all efforts to develop a situational diagnosis of the willingness for interpersonal forgiveness and health-related quality of life in older individuals are justifiable, through the practical application of instruments that can translate this reality, with the purpose of evidencing and supporting the elaboration of innovative interventions in health, especially in mental health.

This study aims to describe and correlate the willingness to forgive (of oneself, others, and uncontrollable situations) with self-perceived health-related quality of life, including the components of quality of physical health and mental health, as well as the variation of these measures according to the sociodemographic characteristics of the study group.

## 2. Method

### 2.1. Participants

Our non-probabilistic convenience sample comprised 30 older individuals from the Iberian Peninsula of Europe. The inclusion criteria were older individuals (65 years and older), residents of the municipality of Porto (Portugal, PT) and Salamanca (Spain, ES), with an education level of at least 4 years education, with preserved cognitive abilities, and who accepted the invitation to freely participate in the study from June 2024 to September 2024.

### 2.2. Cognitive Preservation Criterion

In order not to compromise the results of the tests applied, for the criterion of cognitive ability to participate in the study, a screening for cognitive deficit/dementia was previously conducted, with instruments already validated for the Portuguese and Spanish populations.

The Mini-Mental State Examination (MMSE) [12] instrument was applied for the Portuguese sample, and the cutoff values were differentiated according to participant’s literacy: from 22 points for individuals with up to 2 years of literacy, 24 points for individuals with 3–6 years of literacy, and 27 points for individuals with ≥7 years of literacy [13]. The MMSE is a widely validated and clinically instrument used in Portugal [12,13] that is used for cognitive assessment in the line of research on active and healthy aging at Santa Maria Health School—Escola Superior de Saúde de Santa Maria (ESSSM); this school is a member of the Age-Friendly University Global Network and supported this study.

For the Spanish sample, the researchers selected the Montreal Cognitive Assessment (MOCA) [14], which was previously applied in the selection and evaluation of the “Study protocol for a randomized controlled trial: Effect of an everyday cognition training program on cognitive function, emotional state, frailty and functioning in older individuals without cognitive impairment” [15], and some of the study participants were included in the sample for the present study. The MOCA score has the following variation/interpretations: 16–30 points (without cognitive deficit); 18–25 points (mild cognitive impairment); 10–17 points (moderate cognitive impairment); <10 points (severe cognitive impairment). The cutoff point most used internationally to indicate cognitive impairment is <26 points [14]. Ref. [15] used MOCA as the cutoff point for the Spanish population, being that for mild cognitive impairment, the score was <21 points, and for mild dementia, the score was <20 points [16].

Although we used different cognitive screening instruments in Portugal (MMSE) and Spain (MOCA), both were applied exclusively as inclusion criteria, aiming to exclude participants with moderate or severe cognitive deficits. We recognize that the MOCA has greater sensitivity for mild impairment [14], while the MMSE tends to underestimate these cases. Therefore, this decision represents a methodological limitation of the study, as there may be subtle variations between groups in terms of cognitive reserve.

### 2.3. Data Collection

In Porto (PT), older individuals were invited to participate in the study voluntarily through a dissemination process in the local community, with the support of the Santa Maria Health School—Escola Superior de Saúde de Santa Maria (ESSSM), and in Salamanca (ES), with the support of the Faculty of Nursing and Physiotherapy of the University of Salamanca.

In Porto (PT), the researchers produced and distributed in the local community (local businesses and churches) an Information Folder about the study, with language accessible to older individuals, inviting them to participate in the study, with direct phone number and e-mail contact information for the researchers. After the first contact, a date and time were set for the initial interview, which was conducted individually in an office at the ESSSM, after the participant met all the inclusion criteria and signed the informed consent for participation. In the next step, the evaluation instruments were applied with the guidance of the researchers.

In Salamanca (ES), the invitation was made to the participants in the “Study protocol for a randomized controlled trial: Effect of an everyday cognition training program on cognitive function, emotional state, frailty and functioning in older individuals without cognitive impairment” project [15], during their return for follow-up at the Faculty of Nursing and Physiotherapy, in the second week of September 2024. After their acceptance for participation and evaluation of their compliance with the study inclusion criteria—age, cognitive aspect, and literacy—the same ethical and data collection procedures were conducted. Application of the specific selected instruments was then performed.

### 2.4. Sociodemographic Characterization

For sociodemographic characterization, a questionnaire with 12 questions was applied that collected gender, date of birth, place of birth, marital status, level of education, profession, current employment status, living at home or in an institution (home/residential), how many adults the participant lived with (cohabitation) and who these adults were, whether they received any social support, and whether they received any support from the community or family.

### 2.5. Assessment of Self-Perceived Health-Related Quality of Life

To assess health-related quality of life, through self-perception of health status, the Abbreviated Health Assessment Form, the 12-Item Short-Form Health Survey (SF-12) was used; this has already been translated and validated in several countries in which the initial 36-item short-form of the scale (SF-36—Health Status Assessment) has already been translated, adapted and validated [17,18,19,20]. Since the SF-12 consists of 12 questions, all extracted from the SF-36 questionnaire, it can be self-administered. It summarizes the eight dimensions of the SF-36, maintaining the conceptual model, and has a scope that includes the following domains: physical functionality (two questions), physical limitations in daily activities (two questions), body pain (one question), general health status (one question), vitality (one question), social functioning (one question), emotional limitations in daily activities (two questions), and mental health (two questions). The SF-12 results are represented by a physical component (PCS) score, with a possibility of up to 20 points, a mental component (MCS) score, with a possibility of up to 27 points, and a total score, with a possibility of up to 47 points, which vary percentage-wise in the “optimal standard” from 0 to 100 in each component; the higher the score, the better the self-perception of health status and health-related quality of life [21,22,23,24].

### 2.6. Assessment of Willingness to Forgive

To measure the willingness to forgive, we chose a scale developed by Laura Thompson [25] and her collaborators in 2005, the Heartland Forgiveness Scale (HFS). The HFS is a self-report instrument with 18 items, composed of three subscales of six items each: (a) self-forgiveness (**FOOSE**): Items 1 to 6; (b) forgiveness of others (**FOOSI**): Items 7 to 12; and (c) forgiveness of uncontrollable situations (**FOOT**): Items 13–18. Participants are asked to indicate the degree to which they identify with each sentence using a seven-point Likert scale, with a higher total score reflecting a greater willingness to forgive others, the self, and/or situations, and vice versa [26,27,28,29].

The interpretation of the HFS scale scores should be calculated by using the total values and the values of each dimension or subscale, which for the Total HFS score can vary from 18 to 126 points; a score of 18 to 89 indicates that a person generally or probably does not forgive themselves, others, and uncontrollable situations, and a score of 90 to 126 indicates that a person generally forgives themselves, others, and uncontrollable situations [25].

Regarding the dimensions or subscales of forgiveness, FOOSE (forgiveness of oneself), FOOT (forgiveness of others) or FOOSI (forgiveness of situations that one does not control), a score from 6 to 18, indicates, respectively, that a person usually does not forgive themself, others, and uncontrollable situations; a score from 19 to 29 indicates that a person may or may not forgive themself, other adults, and uncontrollable situations, and a score of 30 to 42 indicates that a person generally forgives themself, other adults, and uncontrollable situations [25].

The Heartland Forgiveness Scale (**HFS**) was applied, which is widely used, as it is the only scale that measures the act of forgiveness of oneself, others, and uncontrollable situations. This scale has been translated several times, even by the authors themselves, and is available free of charge on their website [25,26,27]. Although one study presents interesting findings and suggests that the HFS scale can be applied to the Spanish population, it has not yet been sufficiently tested. However, the authors of this present study chose to apply the structure of the original instrument, as currently, the HFS is the only scale adapted and validated in the Spanish population that allows simultaneous measurement of the dispositional forgiveness of oneself, others, and situations [28].

For the analysis in this study, we chose to group the means of the **Total HFS** results into 18 to 89 points (which indicates that the person generally or probably does not forgive themselves, others, and uncontrollable situations) and 90 to 126 points (which indicates that the person generally forgives themselves, others, and uncontrollable situations). The other subscales, **FOOSE**, **FOOT**, and **FOOSI**, were grouped into 06 to 29 points (which indicates that the person can generally or probably be ruthless with themselves, other adults, and uncontrollable situations, respectively) and 30 to 42 points (which indicates that the person is generally forgiving of themselves, other adults, and uncontrollable situations, respectively).

### 2.7. Statistical Analysis

Statistical analysis of the data was performed using the Statistical Package for the Social Sciences (SPSS), version 29.0J for Windows (IBM Japan, Tokyo, Japan). Student’s *t*-test was used to evaluate the differences in means, and Pearson’s correlation coefficient was used for the correlations, and the significance level was set at 5% (*p* ≤ 0.05). In addition to the *p*-values, effect sizes (Cohen’s d) were computed to assess the magnitude of the differences between groups. For the correlational analyses, Pearson’s r was interpreted according to standard thresholds (small ≥ 0.10, medium ≥ 0.30, large ≥ 0.50).

### 2.8. Ethical Considerations

This study was conducted in accordance with the Declaration of Helsinki and approved by the Ethics and Research Committee of the Santa Maria Health School—Escola Superior de Saúde de Santa Maria (Approval number: CE2024/02) and by the Ethics and Research Committee of the University of Salamanca (Approval number: 1184). Informed consent was obtained from all the study participants.

## 3. Results

### 3.1. Sociodemographic Profile

The study included 30 older individuals from the Iberian Peninsula of Europe: 15 residents of the city of Porto, Portugal, and 15 residents of the city of Salamanca, Spain. All the participants met the criterion of a minimum age of 65 years, literacy ≥ 4 years of education, and preserved cognitive status. The mean age was 72.9 (SD ± 5.23) years, and there were 22 women (73%) and 8 men (27%). In Portugal, for the prior assessment of cognitive status, the MMSE was applied, with a mean score of 28 (SD ± 1.07) points for 4 to 11 years of education, and 30 (SD ± 1.10) points for ≥12 years of education. In Spain, the MOCA was applied, with a mean score of 25 (SD ± 1.79) points for 4 to 11 years of education, and 26 (SD ± 2.59) points for ≥12 years of education. The two participating groups were similar in terms of sociodemographic characteristics, with no statistical differences (Table 1). This result allowed the union of the two groups for the other evaluations.

### 3.2. Overall Assessment of Self-Perceived Health-Related Quality of Life and Willingness to Forgive

As in the sociodemographic profile, the mean results of the scores obtained in the assessments of self-perceived health-related quality of life (SF-12 Total) and willingness to forgive (Total HFS) were similar between the two participating groups, with no statistically significant differences. The means for SF-12 Total were above 70%, indicating a good self-perception of health, with a slight decrease for the mental component summary (MCS) in the group of participants in Portugal, which was 69.5% (SD ± 24.46), but was also without statistical difference (Table 2).

In the general evaluation of willingness to forgive, although without statistical difference from the means, the Total HFS for the group from Portugal had an average score of 86.9 (SD ± 27.39), which indicates that people usually or probably do not forgive themhimselves, others, and uncontrollable situations, while the group from Spain presented with a Total HFS average score of 92.5 (SD ± 18.66), which indicates that people generally forgive themselves, others, and uncontrollable situations. In the dimension or subscale of self-forgiveness (FOOSE), there was no statistical difference, and all the averages were below 30 points, which indicates that people are usually or probably unforgiving of themselves or other people. This result was repeated for the act of forgiveness of uncontrollable situations (FOOSI) for the group in Portugal; this group’s score was 27.7 (SD ± 10.76), but this was higher than for the group in Spain, who scored 30.9 (SD ± 8.64), which indicates that people generally forgive uncontrollable situations. However, there was no statistical difference between the results, the same as for the act of forgiveness of others (FOOT) (Table 2). This result also allowed the union of the two groups for the other evaluations.

### 3.3. Evaluation of Self-Perceived Health-Related Quality of Life and Willingness to Forgive According to Sociodemographic Variables

The means of the self-perceived health-related quality of life scores, SF-12 Total, do not present a statistical difference according to the sociodemographic characteristics of the study group; however, in the components of perceived physical health (PCS) and mental health (MCS), the scores are significantly better for individuals who are still working activity than those who are retired; the PCS for retirees is 72% (SD ± 21.54) and for those with work activity it is 89.5% (SD ± 4.04), with *p* ≤ 0.01, while the MCS for retirees is 68% (SD ± 23.04) and for those with work activity it is 89.3% (SD ± 4.27), with *p* ≤ 0.01. The mental component summary (MCS) also presents a statistical difference according to sex and cohabitation, being higher for men, MCS = 85.1% (SD ± 18.35), than for women, MCS = 65.6% (SD ± 22.16), with *p* = 0.01. Regarding cohabitation, the scores are better for those who live with the family, MCS = 76.4% (SD ± 21.29), than for those who live alone, 62.4% (SD ± 22.9), with *p* = 0.05 (Table 3). The effect size for this difference for MCS was large (Cohen’s d = 0.98), indicating a substantial impact of work engagement on perceived mental health.

Regarding the means for willingness to forgive, there is a statistical difference for total forgiveness (Total HFS), but only according to educational level (years of schooling), this being higher for adults with 12 or more years, Total HFS = 99.2 (SD ± 22.86), than in adults with 4 to 11 years of education, Total HFS = 84.2 (SD ± 22.13), with *p* = 0.02. In this same characteristic, there was also a statistical difference for willingness to forgive the self (FOOSE) and willingness to forgive uncontrollable situations (FOOSI), this being higher in both subscales for individuals with more than 12 years of education. The disposition of self-forgiveness (FOOSE) also varies in statistical significance, according to work activity, being lower for those who are retired than for those who are active. On the other hand, there is a statistical difference for willingness to forgive others (FOOT), but only according to age group, this being higher for adults aged ≥76 years (Table 4). Here, the medium-to-large effect size (Cohen’s d = 0.85) reflects the importance of education in fostering self-forgiveness capacity.

### 3.4. Evaluation of the Variation Self-Perceived Health-Related Quality of Life Scores (SF-12) and Its Physical Health (PCS) and Mental Health (MCS) Components, Stratified by Willingness for Forgiveness (HFS) and Its Different Subscales

The mean scores for the assessment of total self-perceived health-related quality of life (SF-12 Total) and the SP-12 physical health (PCS) and mental health (MCS) components were stratified according to the results for willingness to forgive (Total HFS) and its subscales, self-forgiveness (FOOSE), forgiveness of others (FOOT), and forgiveness of uncontrollable situations (FOOSI). There was a statistically significant variation in self-perceived health-related quality of life (SF-12 Total) for all the willingness to forgive subscales, with the results always being higher when the score for forgiveness indicated that the person generally forgives themself, other adults, and uncontrollable situations. The same variation was observed for the mental health quality component (MCS) in all the willingness to forgive subscales. For the physical health quality (PCS) component, there was only statistical significance in the act of forgiveness of others (FOOT) subscale, this being 81.1% (SD ± 16.33) when the FOOT score indicated that the person generally forgives other adults, and 64.2% (SD ± 23.59) when the FOOT score indicates that the person can generally or probably be ruthless with other adults (Table 5).

### 3.5. Assessment of the Correlation of Self-Perceived Health-Related Quality of Life (SF-12) and Its Physical Health (PCS) and Mental Health (MCS) Components with Willingness for Total Forgiveness (HFS) and Its Different Subscales

In the analysis of the data obtained, it is possible to affirm that there is a moderate positive direct correlation between the perception of self-perceived health-related quality of life (SF-12 Total) and the willingness to forgive (Total FHS), and in the subscales of willingness to forgive others (FOOT) and uncontrollable situations (FOOSI), and there is a weak positive direct correlation for willingness to forgive oneself (FOOSE) (Table 6).

Regarding the correlations for the components of self-perceived health-related quality of life, PCS, and MCS and the willingness for total forgiveness (Total HFS) and its subscales, it is possible to state that in the physical health component (PCS), there is a moderate positive direct correlation with the willingness to forgiveness others (FOOT), and a weak positive direct correlation for the act of forgiveness of uncontrollable situations (FOOSI) and total forgiveness (Total HFS). There is a negligible positive direct correlation for self-forgiveness (FOOSE). However, in the mental health component (MCS), there is a moderate positive direct correlation for the disposition of total forgiveness (Total HFS) and all its subscales (FOOSE, FOOT, and FOOSI) (Table 7).

## 4. Discussion

The results showed a direct positive correlation between willingness to forgive and self-perceived health-related quality of life among older individuals living in Portugal and Spain. These findings are in line with previous published studies that highlight the role of forgiveness in promoting psychological and social health and quality of life, especially in older populations [6,7]. The positive relationships identified between willingness to forgive and health-related quality of life suggest that, mainly, the factors assessed by the different dimensions/subscales can act as protective factors, mitigating symptoms of anxiety and depression, and consequently improving mental health. This is in line with previous results that revealed an association between perceptions of mental health and willingness to forgive [4].

The correlations in this study between forgiveness and health quality are in line with the findings of the meta-analysis published by Davis et al. (2015), which found a moderate positive direct correlation between self-forgiveness (FOOSE) and both physical health (PCS) and mental health (MCS), in particular [29]. Although there is a direct positive correlation between self-forgiveness (FOOSE) and the participants’ mental health in this current study, in the MCS component, this is less expressive when compared to forgiveness of others (FOOT) and forgiveness of uncontrollable situations (FOOSI). These results are consistent with the conclusions of Vismaya et al. (2024) [5], who highlight that self-care and self-acceptance, which are central aspects in the process of self-forgiveness, are determinants for the reduction of self-critical thoughts and ruminations that can exacerbate depressive symptoms in older individuals.

Ingersoll-Dayton and Krause (2010) [8] demonstrated that the practice of self-forgiveness contributes significantly to the reduction of rumination and symptoms of depression, reinforcing the findings of this study. Forgiveness is associated with positive health, and positive self-perceived health outcomes are also considered an important strategy to maximize positive emotions to deal with interpersonal transgressions in older individuals [29].

Regarding the willingness to forgive others (FOOT), this was the only dimension that presented a moderate positive direct correlation with the perception of physical health (PCS). This association can be explained by the ability of forgiveness to reduce chronic stress, which is related to the development of conditions such as hypertension and cardiovascular disease [4]. The release of resentments and the reduction of hostility have been associated with lower cortisol levels and reduced inflammatory responses, as described by Worthington Jr. (2005) [1], highlighting and confirming the potential of forgiveness to positively influence the physical health of older individuals, suggesting that older individuals who forgive tend to have better indicators of physical and mental health [30].

In addition, the results of this current study indicated that the willingness to forgive tends to be higher in individuals with a higher level of education (≥12 years), which suggests that literacy may play a mediating role in the ability to forgive. Previous studies, such as those by López et al. (2021) [7], suggest that literacy facilitates critical reflection and the understanding of emotional processes, including the ability to recognize and deal with negative feelings, as well as understanding how cultural aspects influence attitudes towards forgiveness. This factor is particularly relevant in aging contexts, where adaptation to changes and losses becomes essential for maintaining a superior health-related quality of life [9].

However, it is important to emphasize that no statistically significant differences were found between the groups in Portugal and Spain regarding sociodemographic characteristics, willingness to forgive, and self-perceived health-related quality of life, which may indicate that the cultural and contextual influences on forgiveness are manifested in a similar way in these two countries of Iberian origin. This absence of significant cultural variation is relevant as it suggests that the mechanisms that connect forgiveness to health-related quality of life found in this sample of Spanish and Portuguese elderly people are in line with a World Health Organization report on aging and health [10].

There was variation according to some sociodemographic characteristics—namely, for the self-perceived health-related quality of life mental component summary (MCS)— according to sex (this was better for men), for cohabitation (better for those who live with the family), and for employment status (better for those who still work); and this last characteristic also showed variation for the physical component summary (PCS). Regarding the disposition for total forgiveness, forgiveness of oneself, and forgiveness for uncontrollable situations, they all varied according to educational level (literacy: higher for individuals with 12 or more years of education). The willingness to forgive oneself and others also varied for work activity (better for those who still work), and age group (better for the oldest, ≥76 years old). These results are in line with other studies, which confirm that the capacity for forgiveness, as it is interpersonal, can vary, mainly according to age and gender; for example, older individuals and men forgive themselves more, and this can vary with positive relationships with others, work and family, autonomy, and life projects, and this is better in older individuals who remain active and with social coexistence [31,32].

Despite the relevant contributions, this study has some limitations. The small sample size (*n* = 30) and the choice of a non-probabilistic sample may limit the generalization of the results to the general population. In addition, the data collection was conducted through self-reports, which may introduce social desirability bias, influencing the participants’ responses, although these were mitigated, in part, by ensuring the criterion of preserved cognitive reserve [7]. Future studies may consider larger and more diverse samples, as well as assessment methods that include objective measures of psychological and physical health and quality of life and experimental studies, for example, by using biomarkers of stress, such as salivary cortisol, to correlate with willingness to forgive and self-perceived health-related quality of life, especially in mental health [33].

These limitations, however, do not contradict the recommendation to implement programs focused on forgiveness, which can be a valuable strategy for the promotion of active and healthy aging, offering older individuals emotional instruments to deal with the challenges inherent to the aging process; this should be an approach that integrates, in addition to psychological interventions, social and educational interventions, based on scientific evidence that forgiveness can reduce symptoms of depression and improve emotional and social health and quality of life in aging populations [5,6,7,8]. These results are in line with studies suggesting that interventions focused on promoting forgiveness and the use of adaptive coping strategies can improve the health-related quality of life of older individuals [34,35].

We suggest the following for the future: The development of a multicentric study to test a multimodal intervention program with psychosocial intervention activities, such as cognitive-behavioral therapy (CBT) sessions, that focuses on promoting self-acceptance and rumination reduction and aims to increase the ability to forgive oneself (FOOSE) [4]. Support groups for the elderly, with weekly meetings facilitated by psychologists or specialized therapists to share experiences related to hurt and forgiveness processes, aimed at strengthening forgiveness of others (FOOT), as social interaction is essential to reduce isolation and improve mental health [7]. Emotional education and training, with emotional literacy workshops focused on the development of emotional intelligence, with an emphasis on emotional regulation techniques and mindfulness practices, aimed at improving the ability to accept uncontrollable situations (FOOSI) and highlighting that these interventions can increase resilience and self-efficacy in forgiveness processes [5,36,37]. Education for forgiveness and positive aging, with monthly seminars addressing the benefits of forgiveness for physical (PCS) and mental health (MCS) [1]. Integrated physical activities, such as group walks and yoga, integrating relaxation and breathing exercises, in addition to improving physical health, can have a positive impact on emotional regulation and stress management, facilitate forgiveness processes, and highlight the importance of forgiveness in the context of active aging [38]. Continuous evaluation and monitoring will enable the evaluation of the effectiveness of such programs and the adaptation of interventions to better meet the needs of participants.

## 5. Conclusions

In this study, health-related quality of life was directly and positively correlated with the willingness to forgive in older individuals. It was confirmed that the willingness to forgive and its different dimensions/subscales (forgiveness of oneself, of others, and uncontrollable situations) can act as predictors of positive self-perceived health-related quality of life, including the different components of physical and/or mental health, and vary according to some sociodemographic characteristics, especially for the self-perceived health-related quality of life mental component summary (MCS). The findings of this study also reinforce the relevance of promoting the process of forgiveness among older individuals to improve the health-related quality of life in this population.

## Figures and Tables

**Table 1 geriatrics-10-00090-t001:** Characterization of the sociodemographic profile of the study group.

	Portugal (*n* = 15)		Spain (*n* = 15)		
	*n*	%	*n*	%	*p* *
**SEX**					
Woman	11	73	11	73	0.50
Man	4	27	4	27	
**AGE GROUP**					
65–69	4	27	4	27	
70–74	9	60	3	20	
75–79	1	6.5	5	33.5	0.50
80–84	1	6.5	2	13	
≥85	0.0	0	1	6.5	
**MARITAL STATUS**					
Civilly Married or in “Facto Union”	8	53	8	53	
Divorced	6	40	4	27	0.50
Widower	1	7	3	20	
**EDUCATIONAL LEVEL**					
Basic Education—1st Cycle to 4th year	6	40	5	34	
Basic Education—2nd Cycle to 5th–6th Grade	0	0	2	13	
Basic Education—3rd Cycle to 7th–9th Grade	2	13	2	13	
Secondary Education—10th–12th grade	0	0	2	13	0.40
Higher Education—Bachelor’s degree (3 years)	1	7	0	0	
Higher Education—Bachelor’s degree (4 years)	3	20	2	13	
Higher Education—Master’s degree	2	13	1	7	
Higher Education—PhD	1	7	1	7	
**WORKING STATUS**					
Employee	3	20	0	0	
From Home	1	7	2	13	0.40
Unemployed	0	0	1	7	
Retired	11	73	12	80	
**COHABITATION**					
Alone	6	40	6	40	
Spouse	7	46	7	46	
Spouse and Children	1	7	1	7	0.50
Children and/or Grandchildren	1	7	0	0	
Sister	0	0	1	7	

* Student’s *t*-test.

**Table 2 geriatrics-10-00090-t002:** Mean health-related quality of life scores (SF-12) and the SF-12 physical and mental health components, and the scores for willingness to fully forgive (FHS), self-forgive (FOSE), forgive others (FOOT), and forgive uncontrollable situations (FOOSI).

	Portugal (*n* = 15)	Spain (*n* = 15)	
	Average ± SD	Average ± SD	*p* *
**SF-12 Score Total %**	71.2 ±18.9	73.3 ± 20.77	0.28
Physical component (PCS) %	73.3 ± 18.07	75.3 ± 24.07	0.24
Mental component (MCS) %	69.5 ± 24.46	72.2 ± 21.48	0.32
**FHS**			
**Score Total**	86.9 ± 27.39	92.5 ± 18.66	0.25
FOOSE	29.7 ± 8.97	29.0 ± 7.24	0.41
FOOT	29.5 ± 8.85	32.6 ± 7.3	0.15
FOOSI	27.7 ± 10.76	30.9 ± 8.64	0.18

*** Student’s *t*-test**. **SF-12** (perceived quality of life and health), **PCS** (physical health component), **MCS** (mental health component), **FHS** (willingness to forgive), **FOOSE** (forgiveness of oneself), **FOOT** (forgiveness of the other), **FOOSI** (forgiveness of uncontrollable situations).

**Table 3 geriatrics-10-00090-t003:** Variation of the mean scores for health-related quality of life (SF-12) and its physical health (PCS) and mental health (MCS) components according to the sociodemographic characteristics of the study group.

		Portugal and Spain (*n* = 30)				
	SF-12 Total %		PCS %		MCS %	
	Average ± SD	*p* *	Average ± SD	*p* *	Average ± SD	*p* *
**Sex**						
Woman	68.8 ± 17.75	0.08	73.4 ± 17.17	0.38	65.6 ± 22.16	**0.01**
Man	81.8 ± 22.26		76.9 ± 30.36		85.1 ± 18.35	
**Age group**						
65 to 75 years old	70 ± 20.25	0.14	71.8 ± 22.51	0.08	68.9 ± 23.14	0.22
≥76 years old	78.3 ± 17.16		81.3 ± 12.87		76.1 ± 21.87	
**Marital status**						
Civilly Married/“Facto Union”	74.4 ± 21.81		72.8 ± 27.04	0.33	75.6 ± 22.48	0.11
Widowed or Divorced	69.7 ± 17.03	0.25	76 ± 11.32		65.4 ± 22.44	
**Educational Level**						
4 to 11 years old	70.4 ± 20.13	0.25	72.9 ± 20.97	0.32	68.7 ± 25.02	0.23
≥12 years old	75.4 ± 19		76.6 ± 21.78		74.5 ± 18.41	
**Working Status**						
Retired or Home-Based	69.6 ± 19.66	3.00	72 ± 21.54	**<0.01**	68 ± 23.04	**<0.01**
Working or Unemployed	89.5 ± 3.32		89.5 ± 4.04		89.3 ± 4.27	
**Cohabitation**						
Live Alone	67.8 ± 17.6	0.14	75.6 ± 11.83	0.38	62.4 ± 22.90	**0.05**
Family (husband/wife and/or children)	75.2 ± 20.67		73.4 ± 25.60		76.4 ± 21.29	
**Live**						
Portugal—Urban Area	71.2 ± 18.9	0.28	73.3 ± 18.07	0.24	69.5 ± 24.46	0.32
Spain—Urban Area	73.3 ± 20.77		75.3 ± 24.07		72.2 ± 21.48	

* Student’s *t*-test. SF-12 (perceived health-related quality of life and health), PCS (physical component summary), MCS (mental component summary).

**Table 4 geriatrics-10-00090-t004:** Variation in the mean scores for willingness to forgive (FHS) and its different dimensions, and the scores for the components of health-related quality of life (SF-12), according to the sociodemographic characteristics of the study group.

	Portugal and Spain (*n* = 30)							
	Total FHS		FOOSE		FOOT		FOOSI	
	Average ± SD	*p* *	Average ± SD	*p* *	Average ± SD	*p* *	Average ± SD	*p* *
**Sex**								
Woman	87.8 ± 24.39	0.21	28.3 ± 8.20	0.11	30.5 ± 8.32	0.26	29 ± 10.59	0.36
Man	94.9 ± 20.09		32.1 ± 7.22		32.6 ± 7.85		30.1 ± 7.40	
**Age group**								
65 to 75 years old	88.5 ± 22.86	0.34	29.8 ± 7.42	0.33	29.3 ± 7.64	**0.02**	29.5 ± 9.59	0.42
≥76 years old	92.8 ± 25.47		28.1 ± 9.93		36 ± 7.76		28.6 ± 10.74	
**Marital status**								
Civilly Married/“Facto Union”	91.3 ± 20.79	0.35	29.8 ± 8.17	0.34	31.6 ± 7.49	0.34	29.9 ± 8.11	0.35
Widowed or Divorced	87.9 ± 26.38		28.8 ± 8.11		30.5 ± 9.04		28.6 ± 11.59	
**Educational Level**								
4 to 11 years	84.2 ± 22.13	**0.04**	27 ± 7.21	**0.02**	30.7 ± 8.58	0.36	26.5 ± 9.35	**0.01**
≥12 years	99.2 ± 22.86		33.4 ± 8.03		31.7 ± 7.62		34.1 ± 8.75	
**Working Status**								
Retired or Home-Based	87.5 ± 22.73	0.13	28.2 ± 7.72	**0.02**	30.7 ± 8.11	0.31	28.6 ± 9.75	0.18
Working or Unemployed	104 ± 24.12		37 ± 5.77		33.3 ± 9.07		33.8 ± 9.54	
**Cohabitation**								
Live Alone	83.8 ± 26.43	0.14	27.5 ± 7.99	0.15	29.7 ± 9.55	0.23	26.7 ± 11.41	0.13
Family (husband/wife and/or children)	93.6 ± 20.65		30.6 ± 8.02		32 ± 7.15		31 ± 8.32	
**Live**								
Portugal—Urban Area	86.9 ± 27.39	0.25	29.7 ± 8.97	0.41	29.5 ± 8.85	0.15	27.7 ± 10.76	0.18
Spain—Urban Area	92.5 ± 18.66		29 ± 7.24		32.6 ± 7.3		30.9 ± 8.64	

*** Student’s *t*-test**. **Total FHS** (willingness to forgive); **FOOSE** (forgiveness of oneself), **FOOT** (forgiveness of the other), **FOOSI** (forgiveness of uncontrollable situations).

**Table 5 geriatrics-10-00090-t005:** Variation in the mean health-related quality of life scores (SF-12) and the physical health (PCS) and mental health (MCS) components, stratified by willingness to forgive (Total HFS) and its different subscales.

Portugal and Spain (*n* = 30)						
	SF-12 Total %		PCS %		MCS %	
	Average ± SD	*p* *	Average ± SD	*p* *	Average ± SD	*p* *
**TOTAL FHS**						
18–89 score **	61.1 ± 2073	**0.01**	66.1 ± 24.47	0.08	57.5 ± 24.65	**<0.01**
90–126 score ***	78.9 ± 15.03		78.1 ± 17.06		79.5 ± 16.47	
**FHS—FOOSE**						
06–29 score ****	64.8 ± 21.33	**0.01**	70.5 ± 24.20	0.16	60.9 ± 24.76	**<0.01**
30–42 score *****	79.7 ± 14.74		78.1 ± 17.06		80.8 ± 15.42	
**FHS—FOOT**						
06–29 score ****	60.9 ± 20.72	**<0.01**	64.2 ± 23.59	**0.02**	58.7 ± 24.40	**0.01**
30–42 score *****	79.8 ± 14.93		81.1 ± 16.33		78.9 ± 17.80	
**FHS—FOOSI**						
06–29 score ****	66.1 ± 21.16	**0.03**	64.2 ± 24.97	0.08	64.2 ± 24.97	**0.03**
30–42 score *****	79.1 ± 14.98		79.2 ± 17.70		79.2 ± 16.70	

* **Student’s *t*-test**. **SF-12** (perceived quality of life and health), **PCS** (physical health component), **MCS** (mental health component), **Total FHS** (willingness to forgive), **FOOSE** (forgiveness of oneself), **FOOT** (forgiveness of the other), **FOOSI** (forgiveness of uncontrollable situations). ** A score of 18 to 89 on the **Total HFS** indicates that the person generally or probably does not forgive themselves, others, and uncontrollable situations. *** A score of 90 to 126 on the **Total HFS** indicates that the person usually forgives themselves, others, and uncontrollable situations. **** A score of 6 to 29 on **FOOSE** (forgiveness of oneself), **FOOT** (forgiveness of others), or **FOOSI** (forgiveness of uncontrollable situations) indicates that the person usually or probably may be unforgiving to themselves, other adults, and uncontrollable situations, respectively. ***** A score of 30 to 42 on **FOOSE** (forgiveness of self), **FOOT** (forgiveness of others), or **FOOSI** (forgiveness of uncontrollable situations) indicates that the person forgives themselves, other adults, and uncontrollable situations, respectively.

**Table 6 geriatrics-10-00090-t006:** Pearson’s correlation between self-perceived health-related quality of life and health (SF-12), and willingness for total forgiveness (HFS), and its different subscales.

	(FOOSE)	(FOOT)	(FOOSI)	TOTAL HFS	SF12 TOTAL
	*r* de Pearson *	*r* de Pearson *	*r* de Pearson *	*r* de Pearson *	*r* de Pearson *
(FOOSE)	1				
(FOOT)	0.7	1.0			
(FOOSI)	0.7	0.7	1.0		
(TOTAL HFS)	0.9	0.9	0.9	1.0	
(SF12 TOTAL)	0.4	0.6	0.5	0.5	1

*** Pearson’s correlation coefficient. SF-12** (perception of quality of life and health); **Total FHS** (willingness to forgive); **FOOSE** (forgiveness of oneself); **FOOT** (forgiveness of others); FOOSI (forgiveness of uncontrollable situations).

**Table 7 geriatrics-10-00090-t007:** Pearson’s correlations between the self-perception components of physical health quality and mental health, and the willingness for total forgiveness (FHS) and its different subscales.

	PCS	MCS	FOOSE	FOOT	FOOSI	Total HFS
	*r* de Pearson *	*r* de Pearson *	*r* de Pearson *	*r* de Pearson *	*r* de Pearson *	*r* de Pearson *
**PCS**	1					
**MCS**	0.6	1				
**FOOSE**	0.2	0.5	1			
**FOOT**	0.4	0.6	0.7	1		
**FOOSI**	0.3	0.5	0.7	0.7	1	
**Total HFS**	0.3	0.6	0.9	0.9	0.9	1

*** Pearson’s correlation coefficient**. PCS (perceived quality of life in the physical health component); MCS (perceived quality of life in the mental health component); Total FHS (willingness for forgiveness); FOOSE (forgiveness of oneself); FOOT (forgiveness of others); FOOSI (forgiveness of uncontrollable situations).

## Data Availability

The data are unavailable due to privacy or ethical restrictions; a statement is still required.

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
