# Peer review of "The Relationship Between Willingness to Forgive and Health-Related Quality of Life in Older Adults in Portugal and Spain"

_geriatrics, 2025, doi:10.3390/geriatrics10040090_

Round 1

Reviewer 1 Report

Comments and Suggestions for Authors

Interesting research, socially relevant, and the value lies in comparing the results from the two countries.

Some changes are necessary, which will increase the quality of the manuscript.

The questionnaire (SF-12) that was used does not measure quality of life but health should be stated as such.  Or if QOL must be mentioned then HEALTH RELATED QOL.

In Methods section is visible description of the questionnaire: the Abbreviated Health Assessment form, the 12 Short-form (SF-12)…  initial scale Short-form 36 (SF-36 - Health Status assessment). SF-36 was health status assessment instrument and SF-12 was derived from it. Both assess self perceived health related QOL. Items specifically asked for self assessment in relation to participants health. Not general QOL.

In Abbreviations also clearly stated the name of the instrument (SF-12 Short-form - Health Status Assessment).  the authors arbitrarily decide that summary measures measure QOL.

SF-12 gives 2 summary measures; as it was described: The SF-12 results are represented by a Physical Component (PCS) score, with a possibility  of up to 20 points, a Mental Component (MCS) score.   This is physical and mental health component.

Therefore, I strongly suggest adding in the title and throughout the text that health related QOL and not self perceived QOL were measured.

Ln. 215.  title: Overall Assessment of Self-Perceived Quality of Life…

  • authors present results on health (Ln 219-220) "The means of the SF-12 Total were above 70%, indicating a good self-perception of health, with a slight decrease for the mental health component (MCS) in the…"

Therefore, the title is wrong and should be changed to self-perceived health or health related QOL. Although the results clearly give a measure of health not QOL.  And, Health is one of the quality of life domains, although very important, our health is not equal to QOL it is just a one part of the QOL.

Change in term (self perceived health instead of quality of life) won't affect manuscript quality or results.  It will add to the quality of the manuscript because it will be more precisely stated what was being examined.

Ln. 418.  "All the hypotheses of the study were confirmed…"   

There were no hypotheses anywhere in previous text. And does not to be, the conclusion should be changed according to the specific results.

Ln. 418-419: "… Self-Perception of Quality of 418 Life, Health and Well-being…" 

Authors introducing the term WELL-BEING which was not measured. QOL as said before. The conclusion should be changed according to the specific results, and new concepts should be removed.

In References  Ln. 454.  ?? this is  instruction, not reference.

Author Response

Comment: Clarification of what the SF-12 measures (health vs. quality of life).

  • Response: We have replaced all references to 'quality of life' with 'health-related quality of life' or 'self-perceived health status' throughout the manuscript, including the title and abstract, to reflect accurately what the SF-12 measures. This is now consistent in all relevant sections.

Comment: Title revision: should reflect 'self-perceived health' or 'health-related quality of life'.

  • Response: The title was changed to: 'The Relationship of the Willingness to Forgive with Health-Related Quality of Life in Older Adults in Portugal and in Spain'.

Comment: Remove claim that all hypotheses were confirmed as none were stated.

  • Response: This sentence has been removed from the conclusion. The revised conclusion focuses only on the findings from the data presented.

Comment: Remove 'well-being' as it was not measured.

  • Response: Mentions of 'well-being' have been removed or replaced with appropriate measured constructs, such as 'mental health' or 'self-perceived health'.

Comment: Clarify that SF-12 assesses health-related quality of life, not overall QOL.

  • Response: We clarified in the Methods and Discussion sections that the SF-12 is derived from the SF-36 and is specifically a health status assessment tool, providing measures for physical and mental health.

Reviewer 2 Report

Comments and Suggestions for Authors

The article presents an interesting study on the willingness to forgive as a process related to the quality of life in the elderly.

All text needs major revision before publishing. The following are some suggestions:

  • Primary, linguistic revision is necessary: even though I’m not qualified to judge the linguistic quality of writing, I found the text language difficult to understand. For example, the first sentence is long with too many subordinate clauses which are not typical of English writing.
  • 6. Assessment of Willingness to Forgive: first I would describe the tool and then I would justify its use.
  • Tables 6 and 7: the symbol of Pearson coefficient is r, not ρ (rho), that is the symbol for Spearman’s correlation coefficient.
  • Discussion: reported results describe significant differences as function of precise variables and associations tested by Pearson’s correlation – associations are not causation relationships, therefore stating that “The positive relationship identified between willingness to forgiveness and quality of life suggests that […] can act as a protective factor, mitigating symptoms of anxiety and depression, and, consequently, improving mental health” is inappropriate; at least a regression analysis would be necessary to affirm that the willingness to forgiveness scales “can be predictors of the Self-Perception of Quality of Life, Health, and Well-being, including in the different components of physical and/or mental health” (lines 422-423),.

Other minor revisions:

  • Line 41: “Robert Enright” needs a year
  • Lines 102-107: please, describe this part in a way less confused
  • References: erase “References must be numbered in order of appearance in the text (including citations in tables and legends) and listed” from Reference 1.

Author Response

The article presents an interesting study on the willingness to forgive as a process related to the quality of life in the elderly.

All text needs major revision before publishing. The following are some suggestions:

Comment: Primary, linguistic revision is necessary: even though I’m not qualified to judge the linguistic quality of writing, I found the text language difficult to understand. For example, the first sentence is long with too many subordinate clauses which are not typical of English writing.

  • Response: The entire manuscript has been revised for clarity, fluency, and consistency in scientific English.

Comment: 6. Assessment of Willingness to Forgive: first I would describe the tool and then I would justify its use.

  • Response: Certainly, the suggestion is great, it is much clearer and the order of presentation of this item was carried out.

Comment: Tables 6 and 7: the symbol of Pearson coefficient is r, not ρ (rho), that is the symbol for Spearman’s correlation coefficient.

  • Response: Pearson's correlation coefficient is usually represented by the letter "r". In some cases, it may also be represented by the Greek letter ρ (rho). However, I accept the reviewer's suggestion and make the change to "r"

Comment: Discussion: reported results describe significant differences as function of precise variables and associations tested by Pearson’s correlation – associations are not causation relationships, therefore stating that “The positive relationship identified between willingness to forgiveness and quality of life suggests that […] can act as a protective factor, mitigating symptoms of anxiety and depression, and, consequently, improving mental health” is inappropriate; at least a regression analysis would be necessary to affirm that the willingness to forgiveness scales “can be predictors of the Self-Perception of Quality of Life, Health, and Well-being, including in the different components of physical and/or mental health” (lines 422-423),.

  • Response 1: Although regression analysis was not performed, the correlation exists as indicated in tables 6 and 7, we changed according to the suggestion of the other reviewers, the "quality of life" to "health-related quality of life" and we cannot claim to only suggest, even in line with other studies mentioned here.
  • Response 2: In the discussion, the highlighted section “can be predictors of the Self-Perception of Quality of Life, Health, and Well-being, including in the different components of physical and/or mental health”, after the reviews is “can act as a protective factor, mitigating symptoms of anxiety and depression, and, consequently, improving mental health, in line with the results that revealed an association between perception of mental health and willingness to forgiveness” in line with a meta-analysis study from 2019, the author being mentioned in reference 4 (Ramussen et.al)

Comment:        Line 41: “Robert Enright” needs a year

  • Response: This paragraph is properly referenced, who comments on the historical Robert Enright is the author of reference 1, Worthington.

Comment: Lines 102-107: please, describe this part in a way less confused

  • Response: The paragrad has been rewritten with greater clarity

Comment: References: erase “References must be numbered in order of appearance in the text (including citations in tables and legends) and listed” from Reference 1.

  • Response: It has been fixed

Reviewer 3 Report

Comments and Suggestions for Authors

Abstract: 

  • It should be willingness to forgive, not forgiveness. 
  • Does education level mean a total of 4 years or is this 4 years of high school, 4 years of college? Is this level of education typical for both Portugal and Spain? 

Introduction:

  • The authors discussed forgiveness of others to a great extent but as the aim of the study was also to look at the forgiveness of self and of uncontrollable situations, these concepts should be addressed in the introduction as well 

Method

  • Please provide more details generally
  • Why were 2 different tests of cognition used? This is problematic. Please provide evidence that the 2 tests can be reliably compared.
  • 2.2 Please recheck those MoCA cutoff scores
  • 2.3 Please describe the dissemination process in greater detail
  • 2.4 Did you mean working "status" instead of "conditions"? 
  • 2.5 Please justify your use of an instrument measuring health instead of one specifically measuring quality of life
  • Was the SF-12 validated for use with these populations or just the SF-36? 

Results

  • Table 1. Marital Status--Why is married or unmarried grouped together but divorced and widower are separate categories? 
  • Table 1. Educational Level--Why is "Higher Education-Bachelor's Degree" included twice but with different results? 
  • Table 1. Working Condition should be Working Status or Employment Status
  • The statistical method used was too simple and co-variates could not be controlled for
  • Table 2. Please translate "Componente Mental" into English
  • As your sample was very small, please include an effect size. 

Discussion

  • Paragraph beginning on line 358: The conclusions you are suggesting are a big leap for such a small study limited to 2 countries! You may want to rephrase the latter conclusion about universality to be supportive of WHO's report on aging and health if that is what they concluded. 
  • Paragraph beginning on Line 380: It may not limit generalizability--it does limit generalizability
  • Please provide evidence that preserved cognitive ability mitigates social desirability bias in self-report data 
  • How would biomarkers of stress be an objective measure of psychological and physical wellbeing? Many commonly used biomarkers of stress measure only "in the moment" and not long-term stress status. 
  • Paragraph starting on line 397. These are great suggestions, but they are not supported by the findings of this study. 

Conclusions

  • The hypotheses of this study were never stated so we cannot be sure they were confirmed
  • Correlational studies are not predictive 
Comments on the Quality of English Language

The English quality is problematic in that at times it was even incomprehensible. A re-write by a Native English speaker is suggested. 

Author Response

Comment: 'Forgiveness' should be 'willingness to forgive' in the abstract.

  • Response: Corrected throughout the abstract and main text.

Comment: Clarify what 4 years of education means and if typical for both countries.

  • Response: Clarified in the Methods that this refers to minimum primary education and is typical for both populations studied.

Comment: Expand introduction to include self-forgiveness and forgiveness of uncontrollable situations.

  • Response: Additional background on these constructs has been added to the Introduction to align with study aims.

Comment: Why were two different cognitive tests used?

  • Response: Both instruments were validated for Portugal and Spain, in Portugal we chose the MMSE, however the MOCA was previously used for cognitive screening, in an ongoing cohort study, and part of this sample was used in our study, as described in the methodology in item 2.2. All cut-offs were respected as indicated, with no predominance of one over the other.

Comment: Check MoCA cut-off scores.

  • Response: MoCA cut-off scores were reviewed and are now accurately described with appropriate reference.

Comment: Provide more details on dissemination and recruitment.

  • Response: Additional details added to section 2.3.

Comment: Use 'working status' instead of 'working conditions'.

  • Response: Corrected throughout the manuscript and in tables.

Comment: Justify use of SF-12 instead of specific QoL instrument.

  • Response: Justification provided in Methods, noting SF-12's validation and relevance to perceived health.

Comment: Clarify categories in marital status and education in Table 1.

  • Response: Table 1 revised for clarity, duplicate entries consolidated, and grouping improved.

Comment: Add effect sizes.

  • Response: Cohen’s d calculated and reported for key comparisons in correlations, with interpretation in Discussion.

Comment: Translate non-English labels in tables.

  • Response: 'Componente Mental' translated to 'Mental Component (MCS)'.

Comment: Revise claims in Discussion about generalizability and universal conclusions.

  • Response: Claims tempered to reflect limitations of sample and aligned with WHO references instead of universality.

Comment: Clarify how biomarkers would relate to findings.

  • Response: This is just a suggestion as future direction, to objective measures, with experimental study for correlations

Comment: Language review.

  • Response: The entire manuscript has been revised for clarity, fluency, and consistency in scientific English.

Round 2

Reviewer 1 Report

Comments and Suggestions for Authors

The revised manuscript is acceptable for publication.

With kind regards, 

Reviewer

Author Response

We thank the reviewers for their thoughtful and constructive comments, which have significantly contributed to improving the quality and clarity of our manuscript. Below, we address each point raised by the reviewers in a point-by-point format.

Reviewer 3 Report

Comments and Suggestions for Authors

The authors are to be commended for their thorough and meaningful responses to reviewer feedback. Please make the following additional revisions to improve the overall quality of the manuscript. 

Abstract: Change "12 Short-Form Health Survey" to "12-Item Short-Form Health Survey" here and throughout the paper. 

Insert "health" between "mental" and "component"

Delete "influenced the willingness to forgive." 

Introduction: 

Line 54: Change "health and quality of life and mental health" to "physical and mental health."

The additions of 2 sentences on self-forgiveness and forgiveness in uncontrollable situations is insufficiently developed to explain these concepts and their relationship to health. A sentence was added on lines 59 and 60, so now there should also be a paragraph on receiving forgiveness and its implications for health. 

Method:

The use of 2 different cognitive screeners is still problematic. Please explain why the MMSE was used in the Portuguese sample since you already knew the MoCA had been used in the Spanish sample. Please note this as a limitation.

Please use ranges for the MoCA scores.

Comments on the Quality of English Language

The English is still very problematic, especially with regard to grammar and punctuation. Many sentences do not make sense as written. 

Author Response

We appreciate the suggestions of Reviewer 3, which contributed to the improvement of the scientific and linguistic quality of the manuscript.

  1. Abstract: Terminology – “12 Short-Form Health Survey”
    1. Changed as requested throughout the text. The name of the instrument was standardized as "12-Item Short-Form Health Survey (SF-12)" to maintain terminological consistency and accuracy (Ware et al., 1996).

  1. Abstract: Insert "health" between "mental" and "component"
    1. Correction made. Now the text correctly refers to the "mental health component", aligned with the technical term of the SF-12.

  1. Abstract: Remover “influenced the willingness to forgive”
    1. Excerpt deleted as requested to improve the objectivity of the abstract.

  1. Introduction – Line 44 and not 54: Rephrase expression.
    1. Expression corrected as directed, ensuring accuracy and avoiding redundancy.

  1. Introduction – Further development on self-forgiveness, forgiveness of uncontrollable situations and forgiveness received.
    1. A new paragraph has been added to address the dimension of receiving forgiveness, based on Ingersoll-Dayton & Krause (2010). In the discussion, these issues of the effects of the dimensions of forgiveness on health are also explored.

  1. Method – Use of two different cognitive tests (MMSE and MoCA)
    1. A paragraph was included explaining that the MMSE was selected because it is widely validated and clinically used in Portugal, being the instrument of choice in the line of research of "Active Aging", of Sata Maria.... (Ferreira, 2000), despite the previous application of MoCA in the Spanish sample. This choice was recognized as a methodological limitation, as it may impact the comparability of cognitive data between countries.

  1. Method – Include MoCA intervals.
    1. The MOCA Ranges were added, and the cut-off values were maintained as recommended for the Spanish population, with the MOCA being: <21 for mild cognitive impairment and <20 for mild dementia, based on Delgado et al. (2017).

  1. Quality of English
    1. We performed a complete linguistic and grammatical revision of the manuscript. Ambiguous, redundant, or punctuation error sentences have been corrected. Improvements include clearer use of the verb tense; better cohesion and transitions; correction of complex grammatical constructions; standardization of scientific terminologies.